# Sulphated Polysaccharide from *Acanthophora spicifera* Induced *Hevea brasiliensis* Defense Responses Against *Phytophthora palmivora* Infection

**DOI:** 10.3390/plants8030073

**Published:** 2019-03-22

**Authors:** Sittiporn Pettongkhao, Abdulmuhaimin Bilanglod, Khemmikar Khompatara, Nunta Churngchow

**Affiliations:** 1Department of Biochemistry, Faculty of Science, Prince of Songkla University, Hat-Yai, Songkhla 90112, Thailand; sit_ti_pon@windowslive.com (S.P.); Ameen094584@hotmail.com (A.B.); 2Office of Agricultural Research and Development Region 8, Department of Agriculture, Ministry of Agriculture and Cooperatives, Hat-Yai, Songkhla 90110, Thailand; kjoy2000@hotmail.com

**Keywords:** *Acanthophora spicifera*, λ-carrageenan, *Hevea brasiliensis*, induced resistance, *Phytophthora palmivora*, sulphated polysaccharide

## Abstract

Elicitors from seaweeds are considered an alternative stimulant of plant defenses against pathogenic infection. Finding new sources of elicitors and exploring their effects on plant defenses is a significant undertaking. In this study, we extracted crude polysaccharide (CPS) from *Acanthophora spicifera* (a red alga) and tested the effects of the compound on rubber tree (*Hevea brasiliensis*) defense responses. Accumulations of salicylic acid (SA) and scopoletin (Scp) were measured by HPLC. The expression of SA- and Jasmonic acid (JA)-responsive genes was analyzed by semi-qRT-PCR. Strong anion exchange chromatography and Fourier-transform infrared (FTIR) spectroscopy were used for purification and functional characterization of CPS, respectively. The extracted CPS enhanced rubber tree defenses against *Phytophthora*
*palmivora* infection. It induced SA and Scp accumulations and SA-responsive gene expression, but suppressed JA-responsive gene expression. We successfully separated the non-sulphated polysaccharide (F1) from the sulphated polysaccharides (SPS). Both peaks of SPS (F2 and F3) were identified as lambda (λ)-carrageenan. The F3 fraction showed greater elicitor activity on tobacco leaves. It induced SA and Scp accumulations and peroxidase activity but suppressed catalase activity. Furthermore, the purified λ-carrageenan did not cause cell death in tobacco or rubber tree leaves. Therefore, the elicitor from *A. spicifera* could be an alternative plant stimulant.

## 1. Introduction

*Hevea brasiliensis* (Wild.) Muell.-Arg, or the Para rubber tree, is one of the most important economic crops in Thailand. Its products are exported worldwide, producing significant revenue. *Phytophthora palmivora* infects 1000 or more plant species, including ornamental, horticultural, and agricultural crops. It has been identified as a causal agent of leaf fall and black stripe that destroy the rubber plant [1]. These diseases reduce the quality and yield of rubber latex which presents important problems to cultivators.

Plants establish a state of enhanced defensive capacity in response to appropriate stimuli. This state is called induced resistance which is highly effective against a broad range of subsequent pathogen challenges [2]. Two forms of induced resistance are systemic acquired resistance (SAR) and induced systemic resistance (ISR). SAR involves accumulations of pathogenesis-related proteins and salicylic acid (SA) [3], whereas ISR depends on jasmonic acid (JA) and ethylene (ET) pathways [4]. Many marker genes, such as pathogenesis-related genes 1 (*PR-1*), endo-1,3-β-D-glucanase (*PR-2*), and thaumatin-like protein (*PR-5*), are frequently used to illustrate an SA-mediated plant defense response. On the other hand, genes associated with the JA signaling pathway are chitinase (*PR-3*), hevein-like protein (*PR-4*), and protease inhibitor (*PR-6*) [5].

Elicitors are biofactors or chemicals from various sources that are able to induce physiological changes in targeted living organisms. Elicitors are obtained from many sources and induce morphological and physiological alterations and phytoalexin accumulation in plants. Biotic elicitors are derived from bacteria, fungi, plant cell components, viruses, or herbivores, as well as chemicals that are released by plants at the attack site after infection, whereas abiotic elicitors are metal ions and organic compounds [6]. Poly- or oligosaccharides, such as chitin, xyloglucans, chitosan, β-glucan, and oligogalacturonide exhibit elicitor activity in different plant species and strongly induce phytoalexins in plants [6]. Therefore, elicitors have been considered as alternative tools for disease control in crops.

*Phytophthora* pathogens also produce elicitors. Plants pretreated with these elicitors establish induced resistance. Elicitin (10 kDa) and a new 75 kDa elicitor from *P. palmivora* can induce peroxidase activity and accumulations of scopoletin (Scp) and phenolic compounds both in leaves and a suspension of rubber tree cells [7]. The OPEL from *Phytophthora parasitica* triggers callose deposition and expressions of SA-responsive genes including *PR-1* and *PR-5*. *Nicotiana benthamiana* leaves pretreated with the OPEL before being challenged with *P. parasitica* show disease resistance against the infection [8]. Pretreatment of tobacco leaves with a 34 kDa glycoprotein elicitor (CBEL) from *P. parasitica* causes necrosis, induces defense gene expression, and disease resistance [9]. However, these elicitors require many steps to purify, and tend to induce cell death in plants, so they are not suitable to protect field crops.

Marine macroalgae are considered important alternative sources of polysaccharide elicitors because they do not cause cell death on plant tissues [10,11,12]. Sourced from marine macroalgae and green, brown, and red seaweed, cell wall and storage polysaccharides such as alginates, carrageenans, fucans, laminarin, and ulvans can trigger plant defense responses and enhance resistance to viral, fungal, and bacterial infections [13]. For example, sulphated carrageenan from the red alga, *Hypnea musciformis*, induces phytoalexin accumulation, expressions of genes involved in SA and JA signaling pathways, and results in resistance against tobacco mosaic virus infection [12]. Glucuronan and ulvans from the green alga, *Ulva lactuca*, induce SA and phenolic compound accumulations in tomato seedlings and also significantly induce resistance in tomatoes to *Fusarium oxysporum* infection [14]. A linear β-1,3-glucan extracted from the brown alga, *Laminaria digitata*, induces expressions of *PR-1*, *PR-2*, *PR-3,* and *PR-5* genes [10]. In addition, ulvans and oligoulvans from *U. lactuca* increase catalase, peroxidase, and polyphenol oxidase activities and induce phenolic compounds and lignin contents in apples, which lead to resistance against *Penicillium expansum* and *Botrytis cinerea* infections [15].

*Acanthophora spicifera* is a red seaweed of the family Rhodomelaceae (order Ceramiales). The major polysaccharides of many red macroalgae are carrageenans. The structure of carrageenan from *A. spicifera* is reported to be a λ-type-carrageenan [16]. The λ-carrageenan consists of a repeating dimer of D-galactose (A unit), sulphated in C2, linked to a D-galactose (B unit) sulphated in C2 and C6. Duarte et al. [17] reported that the sulphated polysaccharide from *A. spicifera* consisted of A and B units of galactose molecules. In addition, some of the B unit was 3,6-anhydro-α-L-galactose which resulted in the loss of C6-sulphate. However, an alternative purification method and applications of *A. spicifera* extract in plant protection have rarely been reported.

SPS from diverse red algae induces different defense arsenals in several plant species. Mercier et al. [18] reported that the algal polysaccharide carrageenan, λ-carrageenan, causes SA production and fluorescent compound accumulation in tobacco leaves. *Arabidopsis thaliana* treated with ι- or κ-carrageenans reduces leaf damage caused by *Trichoplusia ni*; in addition, ι-carrageenan induces the expression of defense genes, including *PR-1* and *PDF1.2* [19]. *Arabidopsis* pretreated with λ-carrageenan elevates expression of the JA-related genes, *AOS*, *PDF1.2*, and *PR-3,* and resists *Sclerotinia sclerotiorum* infection. In contrast, the ι-carrageenan increases susceptibility to this pathogen [20]. Carrageenan is not only a beneficial plant inducer that stimulates plant defense responses, but is also a compound that improves the growth of plants [21]. Thus, SPS from *A. spicifera* could be an alternative source of a plant defense inducer.

Since SPS from *A. spicifera* has not yet been reported as a plant bio-stimulant, we investigated the effects of polysaccharide extracted from *A. spicifera* on the rubber tree plant. In this study, we utilized the crude polysaccharide (CPS) of this seaweed as a bio-elicitor to enhance the disease resistance of *H. brasiliensis* against *P. palmivora* infection and measured inductions of SA and Scp, including expressions of *HbPR-1*, *HbGLU*, and *HbASI*. Furthermore, we purified λ-carrageenan from the CPS. The purified λ-carrageenan was also used to test for cell death and responsive enzyme (catalase and peroxidase) induction.

## 2. Results

### 2.1. CPS Induced Resistance in Rubber Tree Leaves.

CPS from *A. spicifera* was precipitated by ethanol for use as a plant elicitor. Leaves were treated with the CPS for 24 h and then infected with zoospores of *P. palmivora*. The infected leaves were classified in five levels of infection (Figure 1A). We found that the CPS reduced the numbers of *P. palmivora*-infected leaves (*P*-value < 0.001) (Figure 1B) and infected leaves showed less severe signs of the disease (Figure 1C) (*P*-value = 0.003). The disease severity of the CPS-treated leaves was lower by about 2-fold than controls which were treated with sterile distilled water. The result revealed that the CPS extracted from *A. spicifera* could act as a plant defense inducer against *P. palmivora* infection on rubber tree leaves.

### 2.2. SA and Scp in Rubber Tree Leaves were Induced by the CPS.

Rubber tree leaves, after being sprayed with the CPS, were analyzed for levels of SA and Scp by HPLC. We found that SA was significantly induced at 48 h post treatment (*P*-value = 0.037) and was slightly elevated again at 96 h post treatment (Figure 2A). Scp was highly increased at 48 h (*P*-value = 0.003) and reached the highest level at 96 h post treatment (*P*-value = 0.005) (Figure 2B). The results showed that CPS caused accumulations of SA and Scp in rubber tree leaves.

### 2.3. CPS Induced Expression of HbPR-1 and HbGLU in Rubber Tree Leaves but Reduced the Expression of HbASI.

As the CPS induced rubber tree defenses via inductions of SA and Scp, we further studied the expression of the pathogenesis-related protein genes, *HbPR-1*, *HbPR-2,* or *HbGLU* and *HbPR-6* or *HbASI*, of this plant. To investigate whether CPS induced defenses through the SA or JA signaling pathway, half of the rubber tree leaf was used for SA and Scp analyses and the other half was used for studying expression of these genes. Compared with gene expression in control leaves, the expression of *HbPR-1* was induced about 3-fold at 24 h post-treatment (*P*-value = 0.03) (Figure 3A).

The *HbGLU* expression was significantly more induced at 24 (*P*-value = 0.03) and 72 (*P*-value = 0.02) hours post treatment by about 3-fold and 2.5-fold, respectively (Figure 3B). On the other hand, rubber tree leaves treated with the CPS showed reduced expression of *HbASI*. The *HbASI* expression was down-regulated at 24 h and significantly decreased at 48 h post-treatment (*P*-value = 0.04) (Figure 3C). These results revealed that the CPS induced *HbPR-1* and *HbGLU* expressions but suppressed *HbASI* expression, and also suggested that CPS induced rubber tree defenses through the SA signaling pathway.

### 2.4. The CPS Consisted of Three Fractions and the Major Fraction was SPS, λ-Carrageenans.

A strong anion exchanger, Q sepharose (GE Healthcare), was used to purify the CPS isolated from *A. spicifera*. We could separate the CPS into three fractions, F1, F2, and F3, which were eluted with 0.4 M, 0.8 M, and 1.6 M NaCl, respectively (Figure 4).

The chromatographic elution profiles of the three fractions demonstrated a complete separation of the fractions (Figure 4). The percentages of F1, F2, and F3 were 24%, 15%, and 61%, respectively. The main fraction, F3, was eluted with 1.6 M NaCl (Figure 4).

FTIR analysis was carried out to classify each fraction as a non-sulphated or sulphated polysaccharide and to identify the types of carrageenan present. The spectra presented a band at 2927 cm^−1^ and a broad and strong absorption band at 3387 cm^−1^ (Figure 5). Šandula et al. [22] suggested that the bands near 2930 and 3405 cm^−1^ represented the stretching of the CH bond and OH group, respectively. The spectra also showed a broad and strong absorption band at 950 to 1200 cm^−1^ (Figure 5), which is assigned to CO vibration [22]. There was no absorption band of amide I (C=O) at 1546 cm^−1^ nor of amide II (NH) at 1649 cm^−1^ [23,24]. Since these bands are characteristic of proteins, the results indicated that all fractions were polysaccharides and did not contain any protein.

Surprisingly, we found an absorption band near 1221 cm^−1^ in fractions F2 and F3 (Figure 5B,C), but not in fraction F1 (Figure 5A). The absorption band between 1210 and 1270 cm^−1^ represents the stretching vibration of the S=O bond in the sulphate group [25]. This result indicated that F1 was non-sulphated while F2 and F3 were sulphated polysaccharide (SPS). Carrageenan types were identified from the characteristic peaks of κ-, ι-, and λ-carrageenan. The spectra of κ- and ι- carrageenan present one important band at 845 cm^−1^ from galactose-4-sulphate [26]. Galactose-2-sulphate and galactose-6-sulphate in λ-carrageenan produce absorption bands near 830 and 820 cm^−1^, respectively [27]. We did not detect the absorption band at 845 cm^−1^, but we found absorption bands near 830 and 820 cm^−1^ in the spectra of F2 and F3 fractions (Figure 5B,C).

Even though the FTIR equipment automatically labeled the absorption bands at 833.24 cm^−1^, an absorption band near 820 cm^−1^ was also present in F2 (Figure 5B). For F3, the absorption band at 819.12 cm^−1^ was automatically labeled while an absorption band near 830 cm^−1^ was also observed (Figure 5C). These results suggested that F2 and F3 fractions were λ-carrageenan showing vice versa signals at 830 and 820 cm^−1^. Thus, the CPS extracted from *A. spicifera* consisted of both non-sulphated and sulphated polysaccharide (SPS) and the SPS was λ-carrageenan.

### 2.5. The F3 Fraction Exhibited the Highest Elicitor Activity in Tobacco Leaves and was a Broad Molecular-Weight λ-Carrageenan.

The elicitor activity of each fraction was tested by infiltration into tobacco leaves which was more practical than spraying onto rubber tree leaves. We found that the F1 fraction did not induce Scp accumulation. However, in tobacco leaves infiltrated with F2 or F3 fractions, the level of Scp was higher at 48 h post infiltration (Figure 6A). In addition, only the F3 fraction was still inducing Scp at 96 h (Figure 6A). The results suggest that the SPS, λ-carrageenan in this study, possessed elicitor activity but the non-sulphated form did not, and the F3 fraction exhibited the highest elicitor activity.

We also performed a gel permeation chromatography (GPC) experiment to study the molecular weight distribution of F3, which exhibited the highest elicitor activity. The chromatogram revealed that this λ-carrageenan was a broad molecular-weight polysaccharide (Figure 6B).

### 2.6. λ-carrageenan Strongly Induced SA and Scp Accumulations in Rubber Tree Leaves.

Fraction F3 (λ-carrageenan) was further evaluated for SA and Scp accumulations in rubber tree leaves as earlier investigated with CPS in Figure 2. We found that the SA level started to increase at 48 h (*P*-value = 0.039) and significant accumulation was present at 72 (*P*-value = 0.001) and 96 (*P*-value = 0.001) hours post treatment (Figure 7A). Scp accumulation also started to increase at 48 (*P*-value < 0.001) and 72 (*P*-value < 0.001) hours and was well sustained at 96 h (*P*-value < 0.001) post treatment (Figure 7B). These results suggest that λ-carrageenan not only triggered Scp synthesis in the tobacco plant, but also induced SA and Scp accumulation in rubber tree leaves, and that λ-carrageenan was the bioactive compound responsible for elicitor activity in the CPS.

### 2.7. λ-Carrageenan Suppressed Catalase Activity, but Highly Activated Peroxidase Activity.

Rubber tree leaves treated with F3, λ-carrageenan, were cut in and half of each leaf was used for SA and Scp analyses while the other half was used for studying the activities of the defense-related enzymes, catalase, and peroxidase. After native polyacrylamide gel electrophoresis (PAGE) and staining, the results revealed that catalase activity in the λ-carrageenan-treated leaves was suppressed for 96 h post treatment whereas catalase activity in control leaves was increased. The suppression of catalase activity was observed at 48, 72, and 96 h post treatment (Figure 8A). In contrast, peroxidase activity was strongly induced at 24 h and was still increasing at 48 and 72 h post treatment (Figure 8B). These results suggested that catalase activity was suppressed by λ-carrageenan while peroxidase activity was increased.

### 2.8. λ-carrageenan did not Cause Cell Death either in Tobacco or Rubber Tree Leaves.

λ-carrageenan was sprayed on rubber tree leaves to determine its toxicity. It did not induce cell death in rubber tree leaves (data not shown). We also tested the toxicity of λ-carrageenan on tobacco leaves. λ-carrageenan was infiltrated into tobacco leaves at concentrations of 0.5, 1, and 2 mg/mL. Cell death did not occur at any of the λ-carrageenan concentrations tested (Figure 8C). The result indicated that λ-carrageenan was safe to use as a plant bio-stimulant.

## 3. Discussion

Many marine-derived polysaccharides, such as alginates, carrageenans, fucans, laminarin, and ulvans, trigger plant defense responses which enhance protection against viral, fungal, and bacterial infections [13]. The results of this study revealed that CPS extracted from *A. spicifera* induced rubber tree resistance to *P. palmivora* by reducing the number of infected leaves and severity of the disease (Figure 1B,C). Sulphated fucan oligosaccharides extracted from the marine brown alga, *Pelvetia canaliculata*, and subsequently degraded with a fucan-degrading hydrolase, are able to stimulate both local and systemic resistance in tobacco leaves against the tobacco mosaic virus [11]. The polysaccharide extracted from a red alga, *H. musciformis*, induces biological activity in tobacco leaves against the tobacco mosaic virus [12]. Ulvans extracted from a mixture of several *Ulva* species (mostly *Ulva armoricana*) induce protection against the fungi *Colletotrichum trifolii* in *Medicago truncatula* [28]. The diameter of the necrotic lesion caused by *Erwinia carotovora* on a tobacco plant previously injected with laminarin from the brown seaweed, *Laminaria digitata*, is reduced [10]. Our results demonstrated that CPS from *A. spicifera* potentially contained an elicitor that induced rubber tree resistance against *P. palmivora* infection.

Interestingly, our results showed that the CPS extracted from *A. spicifera* induced SA accumulation in rubber tree leaves (Figure 2A). Many seaweed extracts induce SA. The sulphated fucan oligosaccharides extracted from *P. canaliculata* induce accumulation of SA that make tobacco plants more resistant to the tobacco mosaic virus [11]. Sulphated laminarin (PS3) induces SA accumulation in tobacco leaves [29]. CPS, in the present study, also increased Scp in rubber tree leaves (Figure 2B). Several plants produce Scp, a coumarin phytoalexin, which is an antimicrobial substance induced by pathogen infection or elicited by abiotic agents. Klarzynski et al. [11] demonstrated that Scp triggered by sulphated fucan oligosaccharides supported resistance in tobacco plants. Tobacco leaves infiltrated with sulphated polysaccharide from *H. musciformis* induce blue fluorescence metabolites which are phytoalexins (Scp and sesquiterpenoid) and derivatives from the phenylpropanoid pathways [12]. In addition, the PS3 induces Scp accumulation in tobacco leaves against the tobacco mosaic virus infection [29].

Expression of the SA-responsive genes, *HbPR-1* and *HbGLU* was induced in the CPS-treated rubber tree leaves; on the other hand, a JA-responsive gene, *HbASI* was significantly suppressed (Figure 3). It has been reported that activation of the SA signaling pathway suppresses the JA signaling pathway [30]. The correlation between gene expression and accumulation of SA suggested that the CPS in this study induced rubber defense responses through the SA signaling pathway. In contrast, the sulphated polysaccharide (SPS) from *H. musciformis* enhances expression of genes involving SA and subsequently activated the JA/ET signaling pathway [12]. *P. palmivora* infection is inhibited in *N. benthamina* that expresses the PR-1 protein; furthermore, the PR-1 protein acts as an antifungal protein by inhibiting zoospore germination [31]. The cell wall of the oomycete mainly consists of β-1,3-glucan [32] which is a substrate of β-1,3-glucanase. Consequently, induction of *HbPR-1* and *HbGLU* by our elicitor could enhance rubber leaf resistance against *P. palmivora* infection.

The CPS from *A. spicifera* was purified using a Q sepharose, and was separated into three fractions (F1, F2, and F3) (Figure 4). We were able to remove a non-sulphated polysaccharide fraction (F1) from sulphated polysaccharide. Both fractions (F2 and F3) displayed absorption bands near 830 and 820 cm^−1^, but the F2 fraction displayed a strong absorption band of galactose-2-sulphate at 833.24 cm^−1^ (Figure 5B), whereas the F3 fraction showed an absorption band of galactose-6-sulphate at 819.12 cm^−1^ (Figure 5C). Duarte et al. [17] reported that the sulphated polysaccharide from *A. spicifera* consisted of A and B units of galactose molecules; in addition, some of the B unit was 3,6-anhydro-α-L-galactose. Consequently, the galactose-6-sulphate would not be present, resulting in a lower anion charge. Basically, the weakest ionic interaction molecule is eluted from the column first; molecules with stronger ionic interactions are eluted with a higher salt concentration. The fraction (F3) that displayed the absorption band at 819.12 cm^−1^ possibly had more galactose-6-sulphate than the F2 fraction. Therefore, F3 had a stronger negative charge than the F2 fraction.

After infiltration into tobacco leaves, the non-sulphated polysaccharide, F1, did not trigger Scp synthesis whereas the sulphated polysaccharide (F2 and F3) or λ-carrageenan did (Figure 6A). The better inducer was the fraction eluted with 1.6 M NaCl (F3) because it possessed more negative charge. Similarly, the highly sulphated λ-carrageenan enhances resistance to *S. sclerotiorum*, resulting in smaller leaf lesions on *Arabidopsis*. In contrast, the ι-carrageenan, which is a lower sulphated carrageenan, induces susceptibility to this pathogen [20]. It has been shown that after chemical sulphation, laminarin became a stronger inducer. Clearly, sulphated laminarin, but not laminarin, causes electrolyte leakage and Scp and SA accumulations. In addition, laminarin triggers expressions of ET-dependent PR proteins whereas sulphated laminarin induces expressions of ET- and SA-dependent PR proteins [29]. Of the two algal polysaccharides, larminarin and carrageenan, only carrageenan efficiently triggers the defense gene expression in tobacco leaves [18]. Mercier et al. [18] suggested that λ-carrageenan containing the highest sulphate content was the most active inducer. Furthermore, when the molecular weight distribution of the F3 fraction was analyzed, we found that it was a broad molecular-weight carrageenan (Figure 6B). This finding supported the report of Necas and Bartosikova [33].

Since the F3 fraction showed the greatest effect and was the major component in the CPS, we further used the F3 fraction in purified form, which was a λ-carrageenan, to confirm the effects on rubber tree defense responses. Our results revealed that λ-carrageenan in the F3 fraction induced SA and Scp accumulations (Figure 7). Both the CPS and the purified F3 induced SA and Scp, so the active compound in the CPS that triggered rubber tree defense responses against *P. palmivora* infection was λ-carrageenan. λ-carrageenan has been reported to have different effects on plants. *Arabidopsis* leaves treated with λ-carrageenan show resistance to *S. sclerotiorum* that is correlated with increasing expression of the JA-responsive genes, *PDF1.2* and *PR-3* [20]. Sangha et al. [19] treated *A. thaliana* leaves with ι- or κ-carrageenan and obtained reductions of leaf damage caused by *Trichoplusia ni*. In contrast, λ- carrageenan-treated leaves show the same leaf damage as the controls. Our result was similar to the result of Mercier et al. [18], who reported that λ-carrageenan (1000 µg/mL) induced the accumulation of SA in tobacco leaves.

The perception of elicitors induces phenomena such as ion flux, medium alkalinization and cytoplasmic acidification, oxidative burst, and reactive oxygen species production [6]. In our study, we focused on the plant defensive enzymes, catalase and peroxidase. These enzymes are major H_2_O_2_ scavengers in plants [34]. Surprisingly, our results revealed that catalase activity was highly suppressed at 48, 72, and 96 h after treatment with λ-carrageenan (Figure 8A). At these time points, the levels of SA were elevated (Figure 7A). SA is able to bind SA-binding protein (SABP) [35]. SABP shows catalase activity which is specifically inhibited by SA [36], therefore, at these time points, increasing SA might bind and inhibit catalase activity (Figure 8A). In contrast, the activity of the other H_2_O_2_ scavenger, peroxidase, was increased (Figure 8B). Hiraga et al. [37] showed that peroxidase played important roles in various physiological processes such as lignification and defense response against pathogen infection. So, induction of peroxidase might promote rubber tree defense responses against *P. palmivora* infection.

The toxicity of λ-carrageenan was tested by infiltrating this elicitor into tobacco leaves. No cell death was detected (Figure 8C), therefore, this sulphated polysaccharide from *A. spicifera* was a friendly stimulant. Similarly, another sulphated polysaccharide, extracted from the red alga *H. musciformis*, do not cause cell death in tobacco leaves [12]. Klarzynski et al. [11] also supported that sulphated oligofucans did not display signs of toxicity on tobacco leaves. Furthermore, carrageenan is not only a beneficial plant inducer to enhance plant defense responses, but has also been shown to promote plant growth [21]. Consequently, the λ-carrageenan from *A. spicifera* could be an alternative elicitor used for enhancing plant defense response.

We successfully used the CPS from *A. spicifera* to induce rubber tree resistance against *P. palmivora* infection via activation of the SA signaling pathway. It has been reported that SA enhances rubber tree defense responses against *P. palmivora* by increasing endogenous SA and Scp contents, lignin formation, peroxidase activity, and expression of *PR-1* [38]. In order to avoid the expense of commercial SA, the CPS from *A. spicifera* could be used as an alternative plant defense stimulant.

## 4. Materials and Methods

### 4.1. Pathogen and Plants

*P. palmivora* isolated from *Hevea brasiliensis* was cultured on potato dextrose agar (PDA) at 25 °C for 1 week. A 5-mm diameter cork-borer was used to cut agar containing *P. palmivora*. The agar plugs were maintained on a V8 agar for 1 week. For zoospore preparation, sterile distilled water was added to a V8 agar plate and further incubated at 4 °C for 15 min. Zoospores of *P. palmivora* were counted with Petroff Hauser under a light microscope. The zoospore concentration at 1x10^5^ zoospores/mL was used in this study.

Rubber tree seedlings (RRIM600 cultivar) at the B2C developmental stage and 8-week tobacco (*Nicotiana tabacum*) plants were maintained in a growth room with photoperiod control (12 h of light and 12 h of dark at 25 °C). *A. spicifera* was kindly provided by the Phetchaburi Coastal Fisheries Research and Development Center, Phetchaburi, Thailand. Salt and sand were removed by washing with tap water. The biomass was air dried at 60 °C to a constant weight, then milled into fine powder.

### 4.2. Crude Polysaccharide (CPS) Extraction

The fine alga powder (5 g) was treated with methanol (85%, 100 mL) and stirred with a mechanical stirrer overnight to remove pigments, lipids, and low molecular-weight carbohydrates. The material was then boiled at 70 °C in 500 mL of distilled water for 4 h with mechanical stirring. The CPS in the mixture was separated from the residue by centrifugation (6500× *g*) for 10 min. The supernatant was poured into three volumes of ethanol (95%) allowing precipitation. The CPS pellet was collected using centrifugation (6500× *g*) for 10 min, dissolved in distilled water, dialyzed, and lyophilized to a constant weight. Before treatment, the CPS was dissolved in sterile distilled water to a concentration of 0.5 mg/mL.

### 4.3. Induction of Local Resistance in Rubber Tree Leaves

The rubber tree leaves were treated with CPS solution at 0.5 mg/mL and sterile distilled water (DW) for 24 h using spraying. After that, the CPS- and DW-treated rubber tree leaves were infected with 1 × 10^5^ zoospores/mL of *P. palmivora* using the same method. At five days after treatment, *P. palmivora* infection was measured by counting the disease severity on infected leaves. The disease severity index (DSI) was determined according to the modified method described by Yang et al. [39]. Five grades of infection symptom were used (grade 0: No symptom; grade 1: Slight symptoms; grade 2: Medium distortion; grade 3: Rather serious distortion; grade 4: Extremely serious symptoms) (Figure 1A) and the DSI was calculated using the following formula.
$$\begin{array}{l} \mathrm{Disease}\;\mathrm{severity}\;\mathrm{index}\;(\mathrm{DSI})\\ \;\;\;\;\;\;=\frac{\sum \left(disease\;grade\times no.\;of\;leaves\;in\;each\;grade\times 100\right)}{\left(total\;no.\;of\;leaves\times highest\;disease\;grade\right)} \end{array}$$

Three replicates were performed. Each replicate consisted of 42 independent leaves from 7 independent rubber trees.

### 4.4. Salicylic Acid (SA) and Scopoletin (Scp) Measurements

The DW- and CPS-treated leaves were collected and extracted by the modified method according to Ederli et al. [40] at designated time points. The treated leaves (0.25 g) were crushed into fine powder in liquid nitrogen in a mortar and then 750 µL of 90% MeOH was added and mixed. The fine mashed solutions were transferred into 1.5 mL Eppendorf tubes and centrifuged at 12,000 rpm for 5 min. The supernatants were collected. Pellets were re-extracted with 500 µl of 90% MeOH and mixed with the collected supernatants. Total supernatants were adjusted with 50% w/v trichloroacetic acid to reach the final concentration of 5% w/v of trichloroacetic acid, passed through a 0.2 µm filter, and stored in a 2 mL HPLC vial at −20 °C. The samples were analyzed using an HPLC apparatus of the Agilent 1100 series equipped with a ZORBAX Eclipse XDB-C18, 4.6 × 150 mm, 5 micron column. The mobile phase consisted of 2 solutions, 0.1% formic acid and acetonitrile. A gradient elution method was used as follows (time in min/percentage acetonitrile): 0–2/20, 2–8.50/20–40, 8.51–10/40, and 10.01–13/40–60. Fluorescence detectors, Ex 294 nm, Em 426 nm, Ex 337 nm, and Em 425 nm were used to analyze the levels of SA and Scp, respectively.

### 4.5. Expression of Defense-Related Genes

#### 4.5.1. RNA Extraction and cDNA Synthesis

The DW- and CPS-treated rubber tree leaves were collected at designated time points and stored in liquid nitrogen. The frozen leaves were ground into fine powder with a mortar and pestle. The RNeasy Plant Mini Kit (Qiagen, Valencia, CA, USA) was used to extract RNA from samples of the ground leaves using the manufacturer’s protocol. The quality and concentration of extracted RNA were determined by agarose gel electrophoresis and spectrophotometry (MaestroGen, Hsinchu, Taiwan), respectively. First-strand cDNA synthesis was performed using SuperScript III Reverse Transcriptase (Invitrogen, Carlsbad, CA, USA) and the remaining RNA was eliminated from the cDNA products with RNaseH (Invitrogen, Carlsbad, CA, USA).

#### 4.5.2. Semi-Quantitative Polymerase Chain Reaction (semi-qPCR)

The expression of defense-related genes, *HbPR-1*, *HbGLU,* and *HbASI*, was analyzed by semi-qPCR. The GenBank accession no. KM514666 [31], AY325498 [41], and KM979450 [42] were used in the design of the specific primers of *HbPR-1*, *HbGLU,* and *HbASI* genes, respectively. A housekeeping gene, the mitosis protein YLS8 gene (*HbMito*) of *H. brasiliensis* (GenBank accession no. HQ323250) [43], was selected as a reference gene. The PCR reaction mixture consisted of EmeraldAmp^®^ PCR Master Mix (Takara, Otsu, Shiga, Japan), 0.25 μM of each gene-specific primer (Table 1), and 4 μL of cDNA template. The reaction was performed with an initial denaturation step at 94 °C for 4 min; followed by 35 cycles (for *HbPR-1* and *HbGLU*), and 40 cycles (for *HbASI*) of denaturing at 94 °C for 1 min; annealing at 60 °C for 30 s; extension at 72 °C for 1 min, and a final elongation step at 72 °C for 10 min. The PCR products were separated by electrophoresis in 2.0% agarose gel, and visualized under a UV transilluminator and photographed by a gel imager using VisionWorks LS software (UVP BioSpectrum^®^ MultiSpectral Imaging System™, Cambridge, UK) to measure densitometry values of the DNA bands.

### 4.6. Carrageenan Purification

Strong anion exchange chromatography with a Q sepharose fast flow column (GE Healthcare) was used to purify the SPS (λ-carrageenan) from the CPS. First, the CPS extract (0.25 mg/mL, 25 mL) was loaded into the Q sepharose fast flow column (25 mL volume of gel) at a continuous flow rate of 0.75 mL/min. After that, the column was washed with 25 mL of distilled water. Bound polysaccharides were eluted with 25 mL of 0.4 M NaCl, 30 mL of 0.8 M NaCl, and 25 mL of 1.6 M NaCl, respectively. The phenol-sulfuric acid method was used to determine the eluted polysaccharides in each fraction (2.5 mL). Each fraction was pooled and further desalted by a PD-10 column (GE Healthcare). The desalted solutions were lyophilized into powder.

### 4.7. Carrageenan Type Identification

Each fraction was determined for functional groups using FTIR. Briefly, one gram of each fraction was ground with spectroscopic grade potassium bromide (KBr) powder, and then pressed into 1 mm pellets for FTIR measurement (wave range of 600 to 4000 cm^−1^) using a Bruker Vertex70 spectrometer.

### 4.8. Gel Permeation Chromatography

The purified carrageenan was dissolved in deionized water and then separated by GPC equipment (Shimadzu/LC-10ADvp using Shodex SB 804 HQ as a column). Deionized water at 40 °C was used as the mobile phase to separate carrageenan at a flow rate of 1.0 mL/min.

### 4.9. Total Protein Extraction

The treated and control leaves (0.5 g fresh weight) at designated time points were ground into fine powders after freezing with liquid nitrogen and homogenized using a chilled mortar and pestle with 1 mL of cold 100 mM Tris-HCl buffer at pH 7.0 containing 0.25% (v/v) Triton-X and 3% (w/v) polyvinylpyrrolidone (PVPP). The homogenates were centrifuged at 12,000 rpm for 20 min at 4 °C. The supernatants were collected and stored at −20 °C for further analysis of enzyme activity.

### 4.10. Antioxidant Enzyme Gel Staining

For native polyacrylamide gel electrophoresis (PAGE), the total proteins extracted from treated leaves were separated by a 4% stacking gel and a 10% separating gel. After 3 h of running at 100 mA, gels were stained with different solutions according to protocols for the detection of catalase and peroxidase activity.

Catalase activity staining was performed according to the method of Woodbury et al. [44]. Briefly, after native PAGE, the gel was immersed in 3 mM H_2_O_2_ for 25 min at room temperature and transferred into the staining solution containing 1% K_3_Fe(CN)_6_ and 1% FeCl_3_ for 4 min. To stain peroxidase activity, we followed the method of Stafford and Bravinder-Bree [45]. The gel was immersed in a staining solution consisting of 50 mM sodium acetate buffer at pH 5.4, 0.05% *ο*-dianisidine, and 0.1 M H_2_O_2_ at room temperature for 30 min and fixed with 50% ethanol.

### 4.11. Toxicity Test

λ-carrageenan at 0.5, 1, and 2 mg/mL and DW were infiltrated into tobacco leaves for cell death induction. At 5 days post infiltration, treated tobacco leaves were photographed with a digital camera.

### 4.12. Statistical Analysis

The presented values are mean values ± standard error (SE) from the results of at least three independent biological replicates. One-way analysis of variance (ANOVA) according to Duncan’s multiple range tests was used to determine significance with *P*-value ≤ 0.05 using SPSS Statistics 17.0 software. Pairwise comparisons were conducted according to Student’s t test at *P*-value ≤ 0.05.

## 5. Conclusions

The effect of CPS from *A. spicifera* on plant defense response was demonstrated. In rubber tree samples, CPS induced a defense response against *P. palmivora* infection through the SA signaling pathway. Accumulations of SA and Scp were induced by both CPS and its purified form, so the crude extract was more practical to examine in the field test. However, the λ-carrageenan purification method used by our study would be useful for specific applications in the future.

## Figures and Tables

**Figure 1 plants-08-00073-f001:**
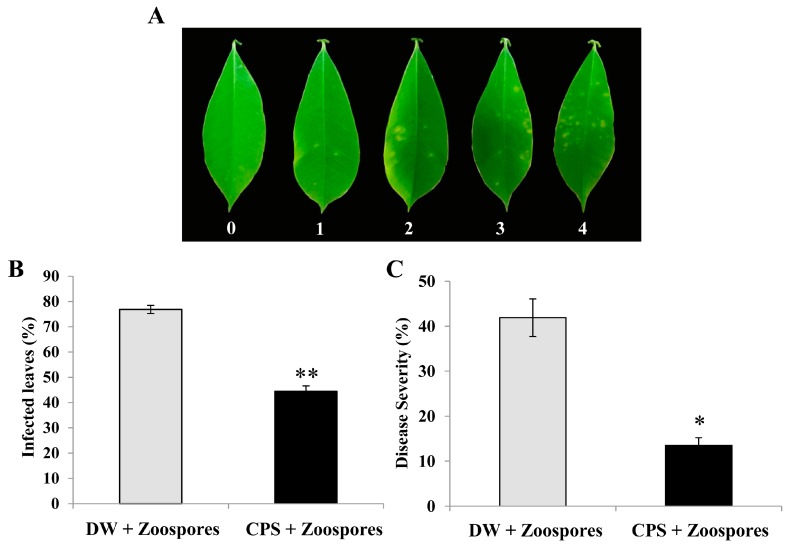
Crude polysaccharide (CPS) reduced infected leaves (%) and disease severity (%) in rubber tree leaves after infection by *P. palmivora*. Five grades of symptom (grade 0: No symptom; grade 1: Slight symptoms; grade 2: Medium distortion; grade 3: Rather serious distortion; grade 4: Extremely serious symptoms) were used to calculate the severity of infection (**A**). The percentage of infected leaves (**B**) and disease severity (%) (**C**) of rubber tree leaves after treatment with 0.5 mg/mL CPS and sterile distilled water (DW) for 24 h and subsequent inoculation with 1 × 10^5^ zoospores/mL of *P. palmivora*. The infected leaves and disease severity were monitored at five days post inoculation. Data bars were the means (± SE) of three replicates. Each replicate consisted of 42 independent leaves from seven independent rubber trees. One and two asterisks (* and **) indicate statistically significant differences of CPS-treated leaves compared with DW-treated leaves (*P*-value < 0.01 and *P*-value < 0.001, respectively) using Student’s t-test.

**Figure 2 plants-08-00073-f002:**
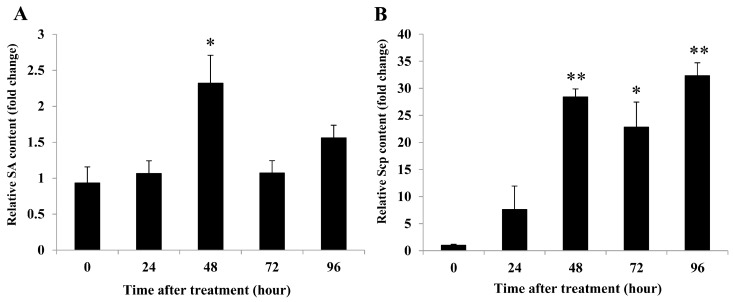
CPS enhanced salicylic acid (SA) and scopoletin (Scp) accumulations in rubber tree leaves. The relative SA (**A**) and Scp (**B**) contents in rubber tree leaves treated with 0.5 mg/mL CPS and sterile distilled water (DW). Data bars were the SA or Scp content (±SE) in CPS-treated leaves relative to DW-treated leaves at each time point. The results derived from three replicates, and each replicate consisted of four plants. One and two asterisks (* and **) indicate statistically significant differences of relative SA or Scp content at each time point compared with relative SA or Scp content at time 0 h (*P*-value < 0.05 and *P*-value < 0.01, respectively) using Student’s t-test.

**Figure 3 plants-08-00073-f003:**
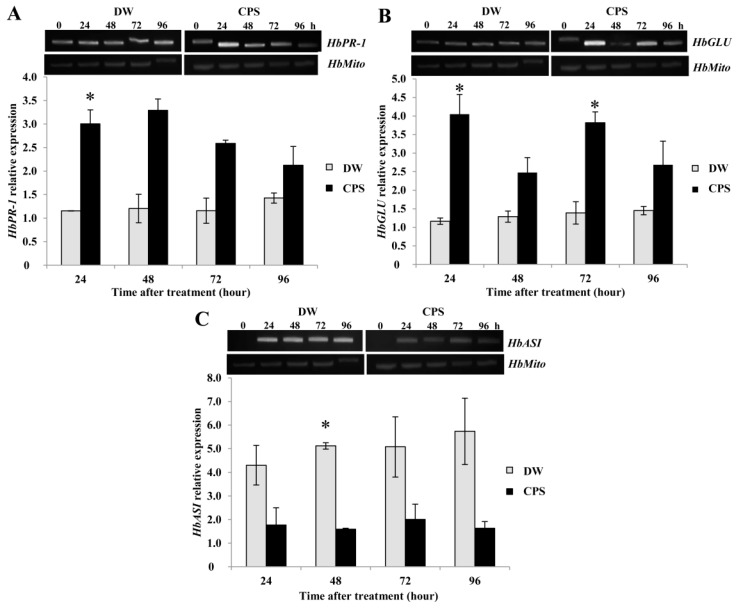
Time course inductions of *HbPR-1* (**A**), *HbGLU* (**B**), and *HbASI* (**C**) expressions by semi-qRT-PCR. The expressions were analyzed on rubber tree leaves treated with 0.5 mg/mL CPS and sterile distilled water (DW). The relative gene expression was calculated in comparison with *HbMito* gene expression. Data bars were the means (±SE) of three replicates, and each replicate consisted of four plants. One asterisk (*) indicates statistically significant differences of CPS-treated leaves compared with DW-treated leaves (*P*-value < 0.05) using Student’s t-test.

**Figure 4 plants-08-00073-f004:**
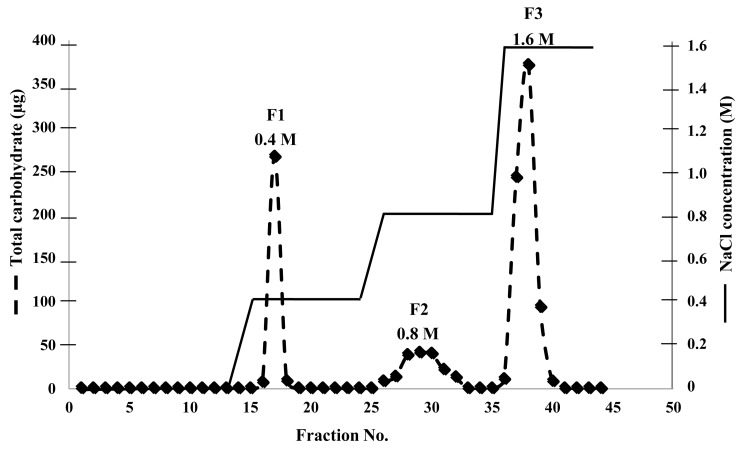
Chromatographic elution profiles of F1, F2, and F3 eluted with 0.4, 0.8, and 1.6 M NaCl, respectively, using a strong anion column, Q sepharose. A total of 2.5 mL of the eluted solution of each fraction were collected and subsequently determined for polysaccharide content.

**Figure 5 plants-08-00073-f005:**
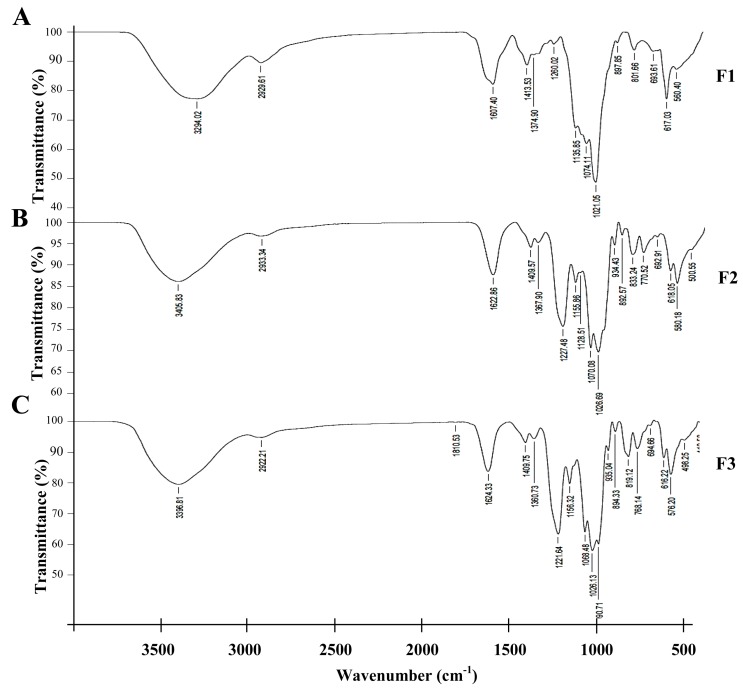
The FTIR spectra of F1 (**A**), F2 (**B**), and F3 (**C**) fractions eluted with 0.4, 0.8, and 1.6 M NaCl, respectively, through a Q sepharose column.

**Figure 6 plants-08-00073-f006:**
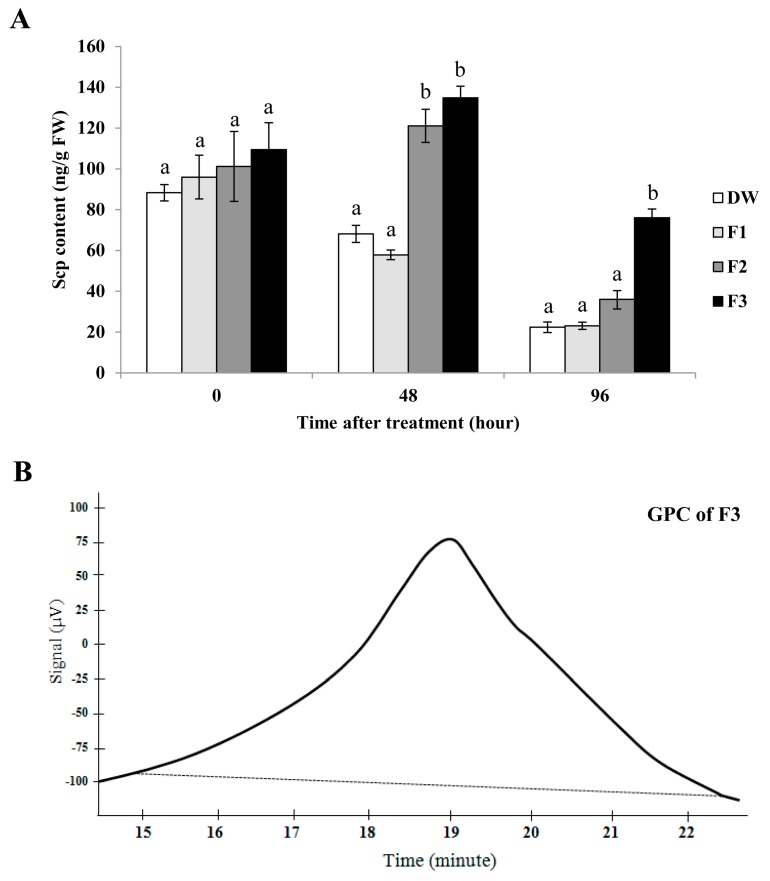
Scp contents in tobacco leaves treated with purified F1, F2, and F3 from CPS of *A. spicifera* and sterile distilled water (DW) (**A**). Data bars indicate the means (±SE) of three replicates. Each replicate consisted of four independent tobacco leaves. The gel permeation chromatography(GPC) elution curve of fraction F3 (**B**). Significant differences utilizing one-way analysis of variance (ANOVA) and Duncan’s multiple range test (*P*-value = 0.05) were presented by differences among treatments.

**Figure 7 plants-08-00073-f007:**
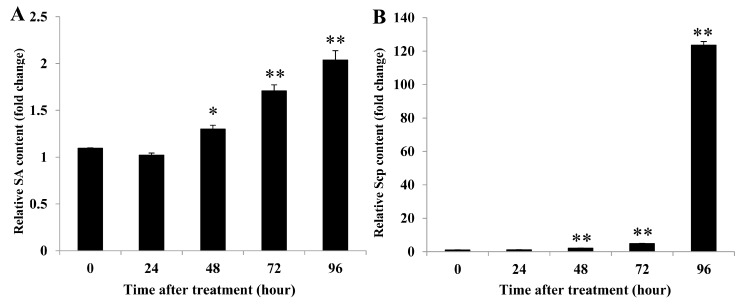
The λ-carrageenan enhanced SA and Scp accumulations in rubber tree leaves. The relative SA (**A**) and Scp (**B**) contents in rubber tree leaves treated with 0.5 mg/mL λ-carrageenan and sterile distilled water (DW). Data bars were the SA or Scp content (±SE) in λ-carrageenan-treated leaves relative to DW-treated leaves at each time point. The results derived from three replicates, and each replicate consisted of four plants. One and two asterisks (* and **) indicate statistically significant differences of relative SA or Scp content at each time point compared with relative SA or Scp content at time 0 h (*P*-value < 0.05 and *P*-value < 0.01, respectively) using Student’s t-test.

**Figure 8 plants-08-00073-f008:**
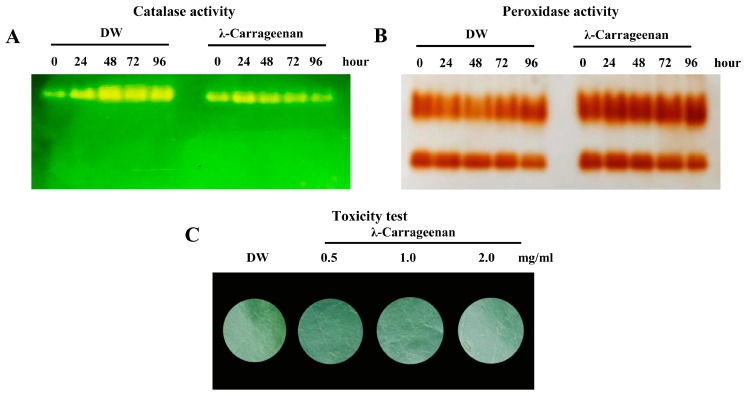
Activity staining of catalase (**A**) and peroxidase (**B**) activity. The rubber tree leaves were treated with 0.5 mg/mL λ-carrageenan and sterile distilled water (DW). Each treatment comprised 12 leaves from different rubber trees. In the toxicity test (**C**), 10 tobacco leaves were infiltrated with 0.5, 1.0, and 2.0 mg/mL λ-carrageenan and sterile distilled water (DW). The symptom induction on tobacco leaves was monitored on day five, post infiltration.

**Table 1 plants-08-00073-t001:** Specific primers used for semi-qRT-PCR.

Gene	Primer Name	Sequence (5ʹ−3ʹ)	Amplicon Size (bp)	GenBank Accession No.
*HbPR1*	*HbPR1*-F	ATGCCCATAACCAAGCACGAGCAG	364	KM514666
	*HbPR1*-R	CCAGGAGGGTCGTAGTTGCATCCA		
*HbGLU*	*HbGLU*-F	GCCTTACCAATCCTTCCAATGC	449	AY325498
	*HbGLU*-R	ATAACCTCGCTGACCATCCCAC		
*HbASI*	*HbASI*-F	GGGCGAAGCCCATATTCACCC	591	KM979450
	*HbASI*-R	CAGAAATCAGAAGCAGACTTCTGCG		
*Hbmitosis protein YLS8*	*HbMito*-F	TGGGCTGTTGATCAGGCAATCTTGGC	577	HQ323250
	*HbMito*-R	TGTCAGATACATTGCTGCACACAAGGC		

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
