# Peer review of "Sulphated Polysaccharide from Acanthophora spicifera Induced Hevea brasiliensis Defense Responses Against Phytophthora palmivora Infection"

_plants, 2019, doi:10.3390/plants8030073_

Round 1
Reviewer 1 Report
This paper describes sulfated polysaccharideselicitors from a new algal source. While the paper is very well written and the design and execution of the experiments appears sound, there is nothing novel about this research. This class of elicitors is very well studied and, other than a new algal source, I don't see what this works adds to our understanding of inducers of disease resistance.
While the elictors do induce all of the expected changes in tobacco and Hevea, the level of resistance induced is relatively modest. It would have been more interesting and more impactful to see if these elicitors provided better protection than others that have been characterized, and how well these elictors perform in a larger field trial with proper comparisons (e.g. other inducers and a standard fungicide).
Author Response
Response to reviewer 1
Point 1: This paper describes sulfated polysaccharides elicitors from a new algal source. While the paper is very well written and the design and execution of the experiments appears sound, there is nothing novel about this research. This class of elicitors is very well studied and, other than a new algal source, I don't see what this works adds to our understanding of inducers of disease resistance.
Response 1: We agree with your comment that our manuscript not having novel issues about class of elicitors which is sulphated polysaccharide from red alga. But this class of elicitor displays various activities in plants and cannot predict the outcome of it. As we discussed that λ-carrageenan has been reported to have different effects on plants. Arabidopsis leaves treated with λ-carrageenan showed resistance to S. sclerotiorum that was correlated with increasing expressions of JA-responsive genes, PDF1.2 and PR-3 [20]. Sangha et al. (2011) treated A. thaliana leaves with ι- or κ-carrageenan and obtained reductions of leaf damage caused by Trichoplusia ni. In contrast, λ- carrageenan-treated leaves showed the same leaf damage as the controls [19]. You can see that λ-carrageenan had various effects depended on plants and pathogens and it induced plants through different signaling pathway. Our results in the manuscript suggested that λ-carrageenan induced rubber tree resistance against P. palmivora infection through SA signaling pathway not JA signaling pathway as reported in Arabidopsis.
We not only presented the induced resistance but the purification method was also reported. Now, we have used this method to purity carrageenan for testing its effect on bacterial inhibition.
Point 2: While the elicitors do induce all of the expected changes in tobacco and Hevea, the level of resistance induced is relatively modest. It would have been more interesting and more impactful to see if these elicitors provided better protection than others that have been characterized, and how well these elicitors perform in a larger field trial with proper comparisons (e.g. other inducers and a standard fungicide).
Response 2: Our laboratory investigated the possible way that could protect rubber tree from the pathogen. As previous reports, we used chitosan and ABA (https://www.sciencedirect.com/science/article/pii/S0885576517302412) and SA (https://www.researchgate.net/publication/326003567_Salicylic_Acid_Induces_Resistance_in_Rubber_Tree_against_Phytophthora_palmivora) as inducers. The disease severities induced by chitosan and ABA were lower than control about 2 folds. Similarly, disease severity induced by SA was lower than control about 2 folds. In this study, the disease severity of the CPS-treated leaves was also lower about 2 folds. In general, the levels of disease severity that we found were lower than controls about 2 folds which were not abundant. However, the advantage of this study was utilizing of the natural product that will be harmless, reduce cost and may up-value of cheap algae which is considered as waste near the shore. Furthermore, carrageenan has also been shown to promote plant growth, therefore, the CPS can be used as an inducer and fertilizer at the same time of application.

Reviewer 2 Report
The article entitled “Sulphated polysaccharide from Acanthophora 3 spicifera induced Hevea brasiliensis defense 4 responses against Phytophthora palmivora infection” by Pettongkhao et al., described the extraction of crude polysaccharide (CPS) from Acanthophora spicifera and it’s effects on defense responses in rubber tree when challenged with Phytophthora palmivora infection. The authors then measured inductions of SA and Scp, along with HbPR-1, HbGLU and HbASI expressions levels. Finally, they purified λ-carrageenan from the CPS and used to test for cell death and responsive enzyme induction when plants were infected with pathogen. The paper can be accepted for publication in Plants but with the following minor changes:
Authors should shorten the introduction. There are lot of generic details that can be curtailed such as lines 49 – 57.
In results section, lines 127 – 129: Authors mentioned that SA was significantly induced at 48 hours post treatment whereas Fig. 2 showed that SA expression remain almost steady till 72 hours whereas level of SA in DW treated samples fell down. Authors should clarify please.
In results section lines 187 – 190: Authors mentioned the absence of absorption band between 1210 and 1270 cm-1 in sample F1 while in Fig. 5 an absorption band was labeled automatically at 31500px-1 then why F1 was considered non-sulphated. Please clarify.
Lines 220 – 224: Authors mentioned that SA level started to increase at 48 hours but it’s not the case in Fig. 7A. Similarly, Scp accumulation does not started to increase at 48 hours as per the Fig. 7B. Authors please clarify.
Lines 236 – 241: Authors should quantify the band intensities for both catalase and peroxidase SDS-PAGE (Fig. 8A and 8B). Similarly, in discussion, lines 339-340 authors should support with quantification of bands
In materials and methods, line 386: Authors should mention if they have infected DW treated leaves with P. palmivora spores. If yes, then why don’t they saw cell death in the control (Fig. 8C). The authors have mentioned treatment of DW + zoospores in Fig. 1 legend though. They need to mention in M&M too. Further, if they have not infected the DW treated leaves with zoospores for cell death experiment then should include that control for the Fig. 8C
Line 453: Authors please mention for how long the isolated protein samples were stored in -20C as they are recommended to be stored at -80C for long term storage
Author Response
Response to reviewer 2
The article entitled “Sulphated polysaccharide from Acanthophora spicifera induced Hevea brasiliensis defense responses against Phytophthora palmivora infection” by Pettongkhao et al., described the extraction of crude polysaccharide (CPS) from Acanthophora spicifera and its effects on defense responses in rubber tree when challenged with Phytophthora palmivora infection. The authors then measured inductions of SA and Scp, along with HbPR-1, HbGLU and HbASI expressions levels. Finally, they purified λ-carrageenan from the CPS and used to test for cell death and responsive enzyme induction when plants were infected with pathogen. The paper can be accepted for publication in Plants but with the following minor changes.
Point 1: Authors should shorten the introduction. There are lot of generic details that can be curtailed such as lines 49 – 57.
Response 1: We have edited as your suggestion.
Point 2: and Point 4: In the results section, lines 127 – 129: Authors mentioned that SA was significantly induced at 48 hours post treatment whereas Fig. 2 showed that SA expression remain almost steady till 72 hours whereas level of SA in DW treated samples fell down. Authors should clarify please.
Lines 220 – 224: Authors mentioned that SA level started to increase at 48 hours but it’s not the case in Fig. 7A. Similarly, Scp accumulation does not started to increase at 48 hours as per the Fig. 7B. Authors please clarify.
Response 2 and 4: We compared the increasing of SA at each time points by comparing with DW-treated leaves at each time point. For example, the SA level in CPS-treated leaves at 48 hours was compared with the SA level in DW-treated leaves at 48 hours. For clearer presentation, we have replotted those results as relative value at each time point and then performed statistical tests of each time point by comparing with the time 0 hour.
Point 3: In results section lines 187 – 190: Authors mentioned the absence of absorption band between 1210 and 1270 cm-1 in sample F1 while in Fig. 5 an absorption band was labeled automatically at 31500px-1 then why F1 was considered non-sulphated. Please clarify.
Response 3: The major indicator for this classification was the appearance of the absorption band between 1210 and 1270 cm-1 and this band usually shows high absorption. In Fig. 5A, we only see a very small peak at 1260 cm-1 which cannot be classified as a sulphated type. In contrast, in Fig. 5B and 5C, we can see highly absorption bands between 1210 and 1270 cm-1 that why we classified these fractions as the sulphated type.
Another indication was the appearance of the absorption band of sulphate group in each Carbon (C) of galactose molecule. As we mentioned that the spectra of κ- and ι- carrageenan present one important band at 845 cm−1 from galactose-4-sulphate [26] while galactose-2-sulphate and galactose-6-sulphate in λ-carrageenan produce absorption bands near 830 and 820 cm−1, respectively [27]. In Fig. 5A, the bands at 845, 830 and 820 cm−1 were not detected which inferred that no sulphate group in F1 fraction.
Furthermore, Fig. 5B (F2) showed the absorption band at 31500px-1 (about 833 cm-1), a characteristic of galactose-2-sulphate, which was displayed higher than the absorption band near 820 cm−1 because F2 possessed more galactose-2-sulphate than F3.
Point 5: Lines 236 – 241: Authors should quantify the band intensities for both catalase and peroxidase SDS-PAGE (Fig. 8A and 8B). Similarly, in discussion, lines 339-340 authors should support with quantification of bands.
Response 5: We have tried to measure the band intensity using the VisionWorks LS software (UVP BioSpectrum® MultiSpectral Imaging System™, Cambridge, UK), but the obtained intensities were not corresponded to the visual detection. Therefore, we cannot address your suggestion for this point.
Point 6: In materials and methods, line 386: Authors should mention if they have infected DW treated leaves with P. palmivora spores. If yes, then why don’t they saw cell death in the control (Fig. 8C). The authors have mentioned treatment of DW + zoospores in Fig. 1 legend though. They need to mention in M&M too. Further, if they have not infected the DW treated leaves with zoospores for cell death experiment then should include that control for the Fig. 8C.
Response 6: We also infected DW-treated leaves with P. palmivora spores and we have edited in the revised manuscript.
The result in Fig. 8C is the toxicity test of λ-carrageenan. We infiltrated tobacco leaves with λ-carrageenan and DW without infection with P. palmivora. No cell death was detected in Fig. 8C indicating that λ-carrageenan does not have toxicity.
Point 7: Line 453: Authors please mention for how long the isolated protein samples were stored in -20C as they are recommended to be stored at -80C for long term storage.
Response 7: We have never kept the extracted samples in -20°C more than 2 weeks. After extraction, we quickly performed analysis because the protein will be lost as your concern.

Reviewer 3 Report
Dear Authors,
The manuscript presents results from testing an elicitor of plant defense against Phytophthora palmivora infections. The novelty of the manuscript lies in using an extract, specifically polysaccharides, from an alga as elicitor. Results on several different aspects such as disease severity, salicylic acid and gene expression are presented. The authors conclude that one fraction (named F3) had the highest potential as an elicitor based on observed reactions of the plants. They further analyse the fraction to identify it as a lambda carrageenan.
Major comments:
Material/Results/Anywhere: I could not find any sample sizes. I might have missed them, in that case I apologize. The only information regarding sample size I found was in figure description 1 (ll 122-123). But even there “3 replicates” are not well explained.
Results/Discussion: There is a general mix of information in the result and discussion section. In the result section there are references to published articles and results are being discussed (e.g. ll140-143 and ll151-153). In the discussion section there are references to figures in the manuscript (e.g. ll299-304). Please separate your own from others results clearly.
L128/figure 2: You describe/interpret that there is a significant induction of SA at 48 h post treatment. By looking at the figure I rather see a significant decrease of SA in DW treated plants at 48h. The levels of SA in CPS treated leaves are on the same level (around 350 ng/g) until 96 hours. Would you mind to elaborate why the DW treated plants “suddenly” change?
L147/figure 3a: “upregulated throughout 96 hours”… the only significance in the figure is marked at 24 hours.
L221/figure 7: Similar to the comment on figure 2, I don’t see the increase of SA from 48 hours, but a drop in the DW treated leaves. At 96 hours there might be an increase. Either reformulate these sections into descriptions relative to DW treated leaves at each time point or perform statistical tests of the lambda-carrageenan treated leaves from one time point to another.
L387: Where the plants still attached to the plant when you treated them? This makes a difference for inducability. Did you test several leaves of one plant or one leaf of several plants?
Minor comments:
L 25: You introduce tobacco leaves without explaining why. How are they connected to the rest of the study? Later in the manuscript (L 204) you mention that it was easier with the tobacco leaves. Could you explain why? And how related / transferable are results from tobacco leaves to rubber tree leaves?
Introduction: The entire introduction is written in past tense. As far as I am aware, results that have previously been published should be described in present tense. For example in line 58: “Phytophthora pathogens also PRODUCE elicitors.” instead of produced. Please check this throughout.
L50: I would prefer “changes in targeted living organisms.”
L50: You write “In plants, elicitors…”. Do you mean for (as in used for plants) or of (as in made of plants)?
L61 (and other places): Sometimes you explain abbreviations and sometimes not. Is there a pattern to it?
Results: Would it be possible to give exact p-values for your statistical tests?
L112/Figure 1C: Does the percentage of disease severity only refer to leaves that did show signs or all leaves (even those in class 0)?
L116: The first line of your figure description is an interpretation. Please change it into only describing what the figure represents.
L 502 and l 505: Different abbreviation for the first names of the same authors.
I hope these comments help to improve the manuscript.
Author Response
Response to reviewer 3
The manuscript presents results from testing an elicitor of plant defense against Phytophthora palmivora infections. The novelty of the manuscript lies in using an extract, specifically polysaccharides, from an alga as elicitor. Results on several different aspects such as disease severity, salicylic acid and gene expression are presented. The authors conclude that one fraction (named F3) had the highest potential as an elicitor based on observed reactions of the plants. They further analyse the fraction to identify it as a lambda carrageenan.
Major comments:
Point 1: Material/Results/Anywhere: I could not find any sample sizes. I might have missed them, in that case I apologize. The only information regarding sample size I found was in figure description 1 (ll 122-123). But even there “3 replicates” are not well explained.
Response 1:
- For Fig. 1 (disease severity), data bars were the means (±SE) of three replicates. Each replicate consisted of 42 independent leaves from 7 independent rubber trees.
- Fig. 2 (SA and Scp measurements), data bars were the means (±SE) of three replicates, each replicate consisted of 4 independent plants.
- Fig. 3 (gene expressions), data bars were the means (±SE) of three replicates, each replicate consisted of 4 independent plants.
- Fig. 6 (Scp measurement in tobacco), data bars indicate the means (±SE) of three replicates, each replicate consisted of 4 independent tobacco leaves.
- Fig. 7 (SA and Scp accumulations), data bars were the means (±SE.) of three replicates, each replicate consisted of 4 plants.
- Fig. 8 (catalase and peroxidase activity), each treatment comprised 12 leaves from different rubber trees (We pooled the other halves of leaves from the sample used for obtaining results of Fig. 7).
- Fig. 8 (Toxicity test), we used ten leaves of tobacco and we did not get any cell death induction.
We have added these explanations in the revised manuscript.
Point 2: Results/Discussion: There is a general mix of information in the result and discussion section. In the result section there are references to published articles and results are being discussed (e.g. ll140-143 and ll151-153). In the discussion section there are references to figures in the manuscript (e.g. ll299-304). Please separate your own from others results clearly.
Response 2: We have edited as your suggestion.
Point 3 and point 5: L128/figure 2: You describe/interpret that there is a significant induction of SA at 48 h post treatment. By looking at the figure I rather see a significant decrease of SA in DW treated plants at 48h. The levels of SA in CPS treated leaves are on the same level (around 350 ng/g) until 96 hours. Would you mind to elaborate why the DW treated plants “suddenly” change?
L221/figure 7: Similar to the comment on figure 2, I don’t see the increase of SA from 48 hours, but a drop in the DW treated leaves. At 96 hours there might be an increase. Either reformulate these sections into descriptions relative to DW treated leaves at each time point or perform statistical tests of the lambda-carrageenan treated leaves from one time point to another.
Response 3 and 5: We used young rubber tree seedlings. One branch of rubber tree seedling consisted of 3 leaves, in this experiment, we used two branches of each plant. The first branch, we collected samples at T0, T24 and T48 hour in first, second and third leaf, respectively. The second branch, we collected T72 and T96 in first and second leaf. Young seedlings for Fig. 2 were planted earlier than the ones for Fig. 7. In general, there are difficult to get homogeneous seedlings for each plantation. No matter what we were careful for young seedlings and leaf selection, it still had variation in each plantation. Therefore, we used the same leaves for Fig. 2 and Fig. 3 and the same leaves for Fig.7 and Fig. 8. According to your comment, we have reformulated the results into relative value at each time point and then performed statistical tests of each time point comparing with time 0 hour.
Point 4: L147/figure 3a: “upregulated throughout 96 hours”… the only significance in the figure is marked at 24 hours.
Response 4: We have addressed this comment by deleting the “upregulated throughout 96 hours”.
Point 6: L387: Where the plants still attached to the plant when you treated them? This makes a difference for inducability. Did you test several leaves of one plant or one leaf of several plants?
Response 6: We used rubber tree seedling. One branch of rubber tree seedling consisted of 3 leaves, in this experiment, we used two homogeneous branches of each plant. In one plant, we have 6 leaves and each treatment consisted of 7 plants. We labeled these branches and sprayed with CPS on each targeted leaf which still attached to the rubber tree seedling. After that, these CPS- and DW-treated leaves were sprayed with zoospore suspension.
Minor comments:
Point 1: L 25: You introduce tobacco leaves without explaining why. How are they connected to the rest of the study? Later in the manuscript (L 204) you mention that it was easier with the tobacco leaves. Could you explain why? And how related / transferable are results from tobacco leaves to rubber tree leaves?
Response 1: The main objective of using tobacco leaves was to identify which fraction possessed elicitor activity. We usually used tobacco leaves to test elicitor activity and many researchers also use them as a model for testing/explaining the effects of something that they interested in. We used the infiltration method with 1 ml needleless syringe to make sure that each fraction entered the tobacco leaves. In each leaf, we infiltrated with DW (control), F1, F2, and F3. It was reliable for comparison the effect of each fraction because these fractions (F1, F2, and F3) induced in the same condition and in the same leaf. On the other hand, rubber tree leaf is covered with thick cutin that why we cannot use infiltration method.
Since CPS highly induced Scp in rubber tree leaves (Fig. 2B), so we selected Scp as an indicator for testing which fraction possessed elicitor activity. We found that sulphated fractions but not non-sulphated fraction induced Scp which suggested that the active compound in the CPS was the sulphated polysaccharide. In addition, the result also suggested that we could use the crude form in field protection of rubber tree not necessary to use the purified form.
Point 2: Introduction: The entire introduction is written in past tense. As far as I am aware, results that have previously been published should be described in present tense. For example in line 58: “Phytophthora pathogens also PRODUCE elicitors.” instead of produced. Please check this throughout.
Response 2: We have edited as your suggestion.
Point 3: L50: I would prefer “changes in targeted living organisms.”
Response 3: We have edited as your suggestion.
Point 4: L50: You write “In plants, elicitors…”. Do you mean for (as in used for plants) or of (as in made of plants)?
Response 4: We mean for (as in used for plants).
Point 5: L61 (and other places): Sometimes you explain abbreviations and sometimes not. Is there a pattern to it?
Response 5: We also concerned about this issue. We try to find the meaning of OPEL protein but we cannot get any explanation. The original and review articles about that also used “OPEL”.
Point 6: Results: Would it be possible to give exact p-values for your statistical tests?
Response 6: We have edited as your suggestion by adding P-value.
Point 7: L112/Figure 1C: Does the percentage of disease severity only refer to leaves that did show signs or all leaves (even those in class 0)?
Response 7: All leaves (even those in class 0) were used to calculate disease severity.
Point 8: L116: The first line of your figure description is an interpretation. Please change it into only describing what the figure represents.
Response 8: We have edited as your suggestion.
Point 9: L 502 and l 505: Different abbreviation for the first names of the same authors.
Response 9: We have edited as your suggestion.

Reviewer 4 Report
In general, the manuscript was well presented. However, I have made some minor revisions (highlighted in yellow) on the MS which has been uploaded. To be consistent, the citation of references should be by numbers and not by dates as I highlighted in the revised MS.
Furthermore, in the Abstract section I have inserted (a red algae) after the scientific name so that the readers have a better idea. On line 368 on Materials & Methods, what was the concentration of zoospore suspension used and was there any correlation between the disease severity and zoospore concentration? On line 264, which green alga(e) was "Ulvans" derived from?

Author Response
Response to reviewer 4
Point1: In general, the manuscript was well presented. However, I have made some minor revisions (highlighted in yellow) on the MS which has been uploaded. To be consistent, the citation of references should be by numbers and not by dates as I highlighted in the revised MS.
Response 1: We have edited as your suggestions.
Point 2: Furthermore, in the Abstract section I have inserted (a red algae) after the scientific name so that the readers have a better idea. On line 368 on Materials & Methods, what was the concentration of zoospore suspension used and was there any correlation between the disease severity and zoospore concentration? On line 264, which green alga(e) was "Ulvans" derived from?
Response 2:
-We have inserted “a red alga” in the revised manuscript.
-The concentration used in this study was 1x105 zoospores/ml. This concentration was the most suitable for using in this study. If the zoospore concentration was decreased lower than we used, the disease severity was very low resulted in not having significant different between DW- and CPS-treated leaves. In contrast, if we increased the concentration of zoospore suspension, we got the high disease severity in both of DW- and CPS-treated leaves and again not having significant different.
-We have added “ulvans extracted from a mixture of several Ulva species (mostly U. armoricana)” in the revised manuscript.
